# Multiple Gene Expression in Cell-Free Protein Synthesis Systems for Reconstructing Bacteriophages and Metabolic Pathways

**DOI:** 10.3390/microorganisms10122477

**Published:** 2022-12-15

**Authors:** Anwesha Purkayastha, Kathirvel Iyappan, Taek Jin Kang

**Affiliations:** Department of Chemical and Biochemical Engineering, Dongguk University, Seoul 04620, Republic of Korea

**Keywords:** cell-free protein synthesis, VLPs, metabolic pathways

## Abstract

As a fast and reliable technology with applications in diverse biological studies, cell-free protein synthesis has become popular in recent decades. The cell-free protein synthesis system can be considered a complex chemical reaction system that is also open to exogenous manipulation, including that which could otherwise potentially harm the cell’s viability. On the other hand, since the technology depends on the cell lysates by which genetic information is transformed into active proteins, the whole system resembles the cell to some extent. These features make cell-free protein synthesis a valuable addition to synthetic biology technologies, expediting the design–build–test–learn cycle of synthetic biology routines. While the system has traditionally been used to synthesize one protein product from one gene addition, recent studies have employed multiple gene products in order to, for example, develop novel bacteriophages, viral particles, or synthetic metabolisms. Thus, we would like to review recent advancements in applying cell-free protein synthesis technology to synthetic biology, with an emphasis on multiple gene expressions.

## 1. Introduction

Biological processes depend on diverse protein functions, most of which are organized within a complex network in all types of cells. Since even a microorganism’s genetic and metabolic networks are complex, the functions of component proteins have often been studied after being recombinantly expressed and purified. The expression of the protein of interest (POI) can then be modulated positively or negatively in the cell to correlate the POI’s in vitro activity to its role in biological processes at the cellular level. As a fast and reliable way of producing POIs, cell-free protein synthesis (CFPS) was developed and utilized mainly in research [1]. The cell-free nature of this technology is well-suited to functional studies of POIs in a high-throughput manner [2], and is free from negative cellular responses to POI expression, which is especially important when POIs are toxic [3,4,5,6,7]. Until the early 2000s, most research on CFPS had focused on studying a single gene product, partly because the yield of the system was low; typically, radioisotopes or Western blots had to be used to detect the POI. However, in many cases, the single gene product was not enough to understand even one step in a biological process. For example, some enzymes, antibodies (or their active fragments), and virus-like particles (VLPs) are assemblies of multiple heterogeneous protein chains. Furthermore, even monomeric enzymes can be taken only as nodes in a metabolic network that consists of multiple gene products. The partial or complete metabolic pathway can be reconstituted by simply mixing individual, cell-free synthesized proteins with or without purification, and this process has been used to accelerate the design–build–test routine of synthetic biology [8,9,10]. Yet, the co-expression of multiple genes in a single reaction can yield unique outcomes from those obtained by merely mixing individually expressed proteins, as was exemplified in the expression of an active yeast tRNA methyltransferase only when two subunits, Trm8 and Trm82, were co-expressed [11]. 

In this paper, we review recent reports on multiple gene expression by CFPS, focusing on the synthesis of viral particles and metabolic pathway reconstruction (Figure 1), given that these two subjects are closely related to synthetic biology. The keys that enable multiple gene expression by CFPS include improved yield and diversified sources of the extract that contains the cellular components for transcription and translation. Both yield improvements and CFPS system diversification have been thoroughly reviewed elsewhere [8,12,13,14,15]; thus, we will briefly introduce our review by covering the historic achievements. It should be mentioned beforehand that yield improvement, especially in the system based on *Escherichia coli*, was made possible partly because the endogenous metabolisms were still active even though the genomic DNA had been silenced in CFPS systems [16]. This implies that diverse metabolic pathways might be utilized to reconstruct the complete synthetic metabolism by expressing just a few of the exogenous genes in CFPS to yield a metabolically-engineered *E. coli* strain [17], which echoes the importance of developing diverse CFPS systems. 

## 2. Recent Advances in CFPS: An Overview

Improvements in the yield of CFPS systems are directly related to the number and amount of proteins synthesized in a single tube that constitutes a virus, VLP, or metabolic pathway. A typical CFPS system consists of a cell lysate prepared from diverse cellular sources; exogenous components such as the building blocks of RNA and proteins, various salts, and a crowding agent; and DNA to direct the protein synthesis (Figure 2). Since many steps related to gene expression, such as transcription, aminoacylation of tRNAs, and mobilization of translational apparatuses, directly or indirectly consume large quantities of ATP, effective and long-term regeneration of ATP has been pursued to improve CFPS yield [18,19,20,21,22,23]. Typically, a rapid decrease in ATP levels and protein synthesis rates are observed [19,20,24,25,26] in conventional CFPS systems, which utilize phosphate donors such as phosphoenolpyruvate (PEP), acetyl phosphate, or creatine phosphate to regenerate ATP [27,28,29,30]. Depending on the nature of the cell used to prepare the CFPS system, adequate use of a dedicated enzyme might be necessary. For example, a CFPS system derived from *E. coli* is supplemented with creatine kinase when creatine phosphate is to be used as a phosphate donor. However, high costs and nonproductive consumption of energy sources due to endogenous phosphatase activities represent considerable drawbacks of using these substrates [31,32]. Reducing nonproductive consumption of energy sources has been attempted by reducing the nonspecific phosphatase activity endogenous to the system; this has been accomplished through immunodepletion from a wheat germ system [33], or by adjusting the growth medium of *E. coli* cells [32], to name a few examples. However, the accumulation of inorganic phosphate (iP) in the system, which negatively impacts the performance of the CFPS system, was found to be unavoidable as long as phosphate donors were used as energy sources. 

The use of intermediates in the glycolytic pathway, rather than high-energy phosphate-containing substrates, to regenerate ATP offered potential solutions to the problems associated with iP accumulation in the CFPS system. Pyruvate [20,22,34], glucose-6-phosphate [16], and even glucose [24] were shown to be capable of replacing traditional energy sources without iP accumulation. This was possible because all glycolytic enzymes necessary for ATP generation were present and active in the cell-free extract. Another glycolytic intermediate, 3-phosphoglycerate (3-PGA), is an interesting energy source due to its prolonged support of the CFPS reaction [35]. Since 3-PGA was apparently more tolerant toward endogenous phosphatases in the *E. coli* CFPS system, its gradual conversion to PEP via glycolysis was key to prolonged protein synthesis, yielding higher protein production that reached the milligram quantity of POI per milliliter reaction. When this feature was combined with the use of maltose or maltodextrin that had been converted into glucose-6-phosphate, utilizing iP released from PEP, even more pronounced yield increases could be obtained, possibly due to the sustained support of ATP regeneration and prevention of iP accumulation [36]. Recent modifications to the system yielded the so-called *E. coli* TX-TL Toolbox 3.0, which could yield up to 4 mg/mL of POI. Major modifications included a change in the culture conditions in which cells were grown, with the temperature at 40 °C instead of 37 °C, and the combined use of maltodextrin and d-ribose for the reaction [37]. The positive effect of culturing *E. coli* cells at an elevated temperature on the performance of CFPS systems was not completely new. A CFPS system prepared using *E. coli* A19 cells grown at 42 °C outperformed that prepared using *E. coli* cells grown at 37 °C, in terms of both protein yields and solubility [38]. However, the growth medium was rich in amino acids, and the CFPS reactions relied on creatine phosphate, which makes it difficult to directly compare results. Nonetheless, it seems evident that *E. coli* cells grown at 42 °C released more protein folding factors than those grown at 37 °C [38].

Apart from the ATP regeneration system, the proteome of the CFPS system can be manipulated to improve system performance. By analyzing polysomes in the CFPS system, Underwood et al. found that both initiation and elongation steps limited system performance to about 70% of the maximal protein synthesis rate [39]. When purified elongation factors were added to the CFPS reactions, the elongation rate, protein synthesis rate, and yield all increased. Notably, the addition of elongation factors to the reaction also increased the initiation rate, suggesting the coupling of initiation and elongation steps during protein synthesis. A similar approach has been applied to a wider spectrum of protein synthesis machinery. Zhang et al. demonstrated a 2.5-fold increase in CFPS performance by supplementing the native system with five purified components: T7 RNA polymerase, translation initiation factor I and II, ribosome recycling factor, and elongation factor Tu [40]. Recently, the CFPS system derived from mixed cell cultures of *E. coli* overexpressing translation factors has been shown to produce POIs with 100–200% increased yields compared to the control system [41]. Proteomic studies further demonstrated that the enhanced protein yield of these systems in which translation factors were overexpressed was due not only to the high concentrations of translation factors in the system, but also to changes in overall cell protein profiles during overexpression, which created a more optimal environment for the CFPS reaction. New possibilities for enhancing CFPS efficiency in combination with enhanced ATP regeneration strategies will likely emerge soon. 

As exemplified in the aforementioned case of ATP regeneration via glycolytic enzymes active in the CFPS system, the active endogenous metabolism in the CFPS system can be utilized to study the function of synthesized POIs by adding exogenous DNA to the reaction system. Similarly, the presence of machinery for post-translational modification can greatly widen the application of CFPS [42,43]. Therefore, our ability to prepare CFPS systems from diverse organisms, including *E. coli*, *Bacillus subtilis, Bacillus megaterium Corynebacterium glutamicum, Vibrio natriegens, Clostridium autoethanogenum, Pseudomonas putida* wheat germ, rabbit reticulocytes, Chinese hamster ovary (CHO) cells, yeasts, and insect cells, to name a few [44,45,46,47,48,49,50,51,52,53,54,55,56,57,58,59,60,61,62,63,64,65,66], opens new possibilities for studying host-specific virus synthesis or metabolic pathways. For example, genes from *Streptomyces* with high GC content could be more efficiently expressed in a CFPS system derived from *Streptomyces lividans* [46,67]. These systems might be used to find and verify new metabolisms of *Streptomyces*, which are a rich source of various valuable natural products [68]. However, typical protein yields remain relatively low (~100 μg/mL reaction). Eukaryotic CFPS systems can be used to reproduce viral gene expression and subsequent genomic replication, which demands host cellular components as well as viral factors [69,70,71]. A CFPS system derived from *Nicotiana tabacum* BY-2 protoplasts was found to enable the translation and replication of the RNA genomes of tomato mosaic virus, brome mosaic virus, and turnip crinkle virus [69]. Using a BY-2 CFPS system that contained mitochondria for efficient energy supply, the protein synthesis yield reached up to 3 mg/mL, which is comparable to that of *E. coli* systems [72,73,74].

## 3. Virus and Virus-like Particle Synthesis

Whether pathogenic to human beings or not, viruses play important roles in many fields of science. They are sources of new synthetic biology tools, such as novel RNA polymerases, transcriptional regulators, or integrases [75]. Pathogenic viruses or their benign relatives may also provide us with active vaccine candidates [76]. Repurposed viruses engineered from authentic ones can even be used for medical purposes [75]. Undoubtedly, all of these applications and the subsequent engineering demand a reliable method for synthesizing the virus of interest. Though viruses can be produced from the relevant host cells, in vitro systems utilizing CFPS methodology have comparative advantages due to their open nature, which enables controlled and fine-tuned synthesis conditions according to specific needs [77]. A report exists on synthesizing whole infectious viruses that can infect humans (encephalomyocarditis virus) using a CFPS system derived from a HeLa cell [78]. Even though the virus is an RNA virus, a DNA genome was successfully used to prime synthesis. Likely due to safety and security issues, however, the synthesis of eukaryotic viruses is rare. Thus, we would like to confine the scope of our review to bacteriophages and noninfectious virus-like particles (VLPs).

The T7 phage was the first whole virus synthesized in the *E. coli* CFPS system [77]. The 40-kbp-long linear double-stranded DNA genome, encoding about 60 genes, was used at 1 nM to prime whole phage synthesis in vitro, yielding 10^8^–10^9^ infectious phage particles per mL of the reaction when the reaction was incubated for 12 h at 29 °C. In this case, 10^9^ PFU (plaque forming units)/mL from the 1 nM genome corresponds to approximately 0.002 phage/genome being used. As with typical CFPS, the phage titer increased as the concentration of the genome used to prime the reaction increased, and eventually plateaued over a certain value. Interestingly, upon the addition of four dNTPs to the CFPS reaction, the phage titer increased up to 200 times. This clearly suggests that the T7 DNA polymerase synthesized in the cell-free environment replicated its own genome. This possibility was further backed by the finding that thioredoxin positively affected phage synthesis in the CFPS system, as the T7 phage is known to be dependent on thioredoxin present in the host for its replication. The same research group was able to increase the phage per genome copy number used for the CFPS reaction to about 2 [36]. The phage titer reached over 10^11^ PFU/mL reaction, largely because of the improved protein-synthesizing activity of the CFPS system and the proper use of polyethylene glycol 8000 (PEG 8000). PEG is known as a molecular crowding agent that supports supramolecular assemblies [79,80]. It is worth noting that, at elevated PEG 8000 concentrations (~3% *v*/*v*), the phage titer increased, but the synthesis of green fluorescent protein was negatively affected. It thus seems that PEG 8000 primarily helped in phage assemblies. Recently, the same group reported an even further improved T7 phage titer that approached 10^13^ PFU/mL reaction using their improved CFPS system [37]. Though they obtained a lower phage titer, the Simmel group also reported the production of fully assembled T7 at 10^8^ PFU/mL in their *E. coli*–based CFPS system, prepared using lysozyme-assisted sonication [81].

A more sophisticated phage, T4, was also successfully synthesized in a CFPS system derived from *E. coli*. With about 290 genes in its 170-kbp-long double-stranded DNA genome, the T4 phage is one of the largest known bacteriophages, and is as large and complex as eukaryotic herpes viruses [82]. The T4 genome extracted from the phage was successfully used to prime phage synthesis, yielding 10^9^ PFU/mL reaction at its optimal concentration [83]. PEG, the molecular crowding agent, also enhanced phage yield, as was the case in T7 systems. However, the effect of PEG was more pronounced, such that an over 10^5^-fold increase in the phage titer was observed in the range of 0.5% to 1.5% (*v*/*v*) in the reaction. In addition, unlike the case of T7, the addition of dNTPs did not result in an increased phage titer, suggesting a lack of replication. It is intriguing that infectious T4 can be synthesized entirely in vitro, as the complexity and size of the T4 genome resemble the minimal genomic properties for self-sustainable cells. This possibility also opens new opportunities for phage studies, because phage particle assembly was possible in essentially cytoplasmic components of *E. coli*, while some believe that the self-assembly of T4 requires access to the inner membrane of the cell [82].

As mentioned above, the synthesis of eukaryotic viruses, both in vivo and in vitro, presents issues regarding safety and security. However, being noninfectious, VLPs can be developed as vaccines or drug/gene carriers [84], and CFPS can be regarded as an alternative to in vivo production. The advantages of using the well-established CFPS system include the rapid production of VLPs with sequence variations at a high level (~0.5 mg/mL in terms of hepatitis B core antigen or 0.1 nmol assembled hepatitis B VLP/mL) in a readily scalable fashion [85,86]. While VLPs themselves might be used as vaccines, they can also be used as templates for surface modifications with several kinds of ligands. The open nature of the CFPS system facilitates the incorporation of unnatural amino acids with moieties for specific chemical reactions, such as azide–alkyne cycloaddition, into the target VLP in a site-specific manner. This advantage of CFPS over cellular systems was exemplified in the work by [87], in which the hepatitis B VLP surface was modified with *Salmonella typhimurium* flagellin.

Even though many VLPs can be assembled from a single species’ proteins, such as MS2 bacteriophage coat protein or hepatitis B core protein, the co-expression of two viral genes can lead to novel VLPs. In an example presented by Smith et al. [88], Qβ VLP was synthesized in the CFPS system derived from *E. coli* by co-expressing its coat protein (CP) and a unique protein called A_2_. A_2_, of bacteriophage Qβ, is known to facilitate infection and inhibit the host enzyme MurA (UDP-*N*-acetylglucosamine enolpyruvyl transferase), which leads to cell lysis and subsequent viral release from those cells. About 180 CP molecules and a single A_2_ molecule were shown to assemble into one VLP through careful manipulation of the concentrations of the plasmids encoding these proteins. In the design of expression cassettes, the golden ratio of plasmids encoding A_2_ and CP was experimentally determined to be 1:5, which is far from the protein ratio of 1:180. The Qβ yield at that ratio reached a reaction of almost 0.1 mg/mL. Interestingly, other plasmid ratios produced varying Qβ yields, as well as a varied number of A_2_ incorporated into the Qβ VLP. Increasing the proportion of the plasmid encoding the A_2_ gene decreased yields and created either less efficient incorporation of A_2_ into the VLP structure or a lack thereof. Since Qβ CP alone can form VLP [85,89,90], the incorporation of A_2_ into the VLP structure could be better studied in the co-expression system. Considering the toxicity of A_2_ to the *E. coli* cell, the CFPS system was the only, but also the ideal, approach for this study.

## 4. Metabolic Pathway Exploration

### 4.1. Early Example of Multiple Gene Co-Expression in CFPS Systems

Reconstruction of a synthetic metabolic pathway comprising three enzymes—glucokinase (*B. subtilis* YqgR), *N*-acetylglucosamine-phosphate mutase (*Saccharomyces cerevisiae* Agm1), and *N*-acetylglucosamine-1-phosphate uridyltransferase (*E. coli* GlmU)—was carried out in an *E. coli*–derived CFPS system [91].This early work on cell-free multiple gene expression for reconstructing a synthetic metabolic pathway successfully demonstrated the synthesis of UDP-*N*-acetylglucosamine from *N*-acetylglucosamine (GlcNAc), ATP, and UTP (Figure 3A). In this study, however, the product titer was about half (1.9 mM) that of the reaction in which cell-free synthesized individual enzymes were mixed to reconstruct the pathway (4.1 mM). The disproportionate expression levels of the three genes, when co-expressed, could explain the reduced product titer; however, no attempt was made to adjust expression levels in that study. Later, the same group reconstructed a synthetic pathway to form UDP-*N*-acetylmuramyl pentapeptide, a cytoplasmic precursor to bacterial peptidoglycan, from GlcNAc [92]. Since this biosynthetic pathway is well conserved in bacteria, its reconstruction in a CFPS system can lead to the development of antibacterial drugs targeting the entire pathway, rather than just a specific protein [93]. The authors divided the pathway comprising six enzymes into two modules, from which two (*murA* and *B*) and four genes (*murC-F*) were co-expressed, respectively, as is shown in Figure 3A. For gene co-expression, PCR-amplified templates were added to the CFPS reaction at the same mass concentration. The concept of controlling the expression level of a specific protein in multiple gene co-expression formats was demonstrated by using antisense oligonucleotides. However, the relationship between modulated enzyme expression levels and metabolic pathway performance was not evaluated in the work. Interestingly, the authors mention that they had to use linear PCR templates in their work, since their preliminary research to co-express all 6 *mur* genes in a CFPS system was not feasible with three plasmids.

### 4.2. Expression Levels of Multiple Genes Can Be Modulated by Controlling the Template Concentrations

Even though they constitute a single enzymatic step rather than a metabolic pathway, the co-expression of PhaC and PhaE of *Synechocystis* sp. PCC 6803 in a CFPS system is intriguing [94]. PhaC and PhaE are subunits of a type III polyhydroxyalkanoate (PHA) synthase PhaCE, which polymerizes (*R*)-3-hydroxybutyrate [95] and is analogous to *Cupriavidus necator* (formerly *Ralstonia eutropha*) PhaC, a type I PHA synthase. In an attempt to form an active PHA synthase, Numata et al. [94] first produced PhaC and PhaE individually in an *E. coli* CFPS system, and then purified them and reconstituted the purified subunits in varying molar ratios. However, none of the tests resulted in measurable PHA synthase activity. One result of note is that the yields of PhaC and PhaE were 7.8 and 61 mg/L, respectively. Since the molecular weights of PhaC and PhaE are similar [95], it is obvious that the expression of PhaE is more efficient than that of PhaC when a similar concentration of templates is used to prime CFPS. The authors next tried to coproduce the two proteins so that they could form an active heterodimer: PhaCE. Their initial attempt to prime the CFPS reaction with equimolar concentrations of two genes resulted in no appreciable quantity of the complex. When a higher molar concentration of DNA for PhaC than that for PhaE (3:1) was used to prime co-expression, a ratio of approximately 1:1 of PhaC:PhaE was achieved, and the active PHA synthase, PhaCE, was produced. This work, again, demonstrates the importance of co-expressing multiple genes in enabling a CFPS system to mimic a living system. More importantly, it also evidences how the expression levels of individual proteins can be controlled by modulating the concentration of DNA inputs in the system.

The active tripartite nonhemolytic enterotoxin (Nhe) of *Bacillus cereus* was prepared and characterized in a CFPS system derived from Chinese hamster ovary (CHO) cells [96]. Though its name suggests otherwise, Nhe does have hemolytic activity when its three subunits (NheA, NheB, and NheC) possess a specific order and molar ratio [97,98]. Ramm et al. reconstituted active Nhe by co-expressing three genes cloned separately in the same plasmid backbone [96]. When the authors prepared the reconstituted Nhe using molar ratios of three plasmids encoding NheA, NheB, and NheC at 1:1:1 in the CFPS, they could not detect any appreciable hemolytic activity, even though all subunits were properly expressed. Knowing that the optimal molar ratio of NheA:NheB:NheC was 10:10:1 for reconstructing the active Nhe complex [97], the authors changed the plasmid ratio from 10:10:10 to 10:10:1 in a series of experiments aimed at suppressing the expression level of *nheC*; in doing so, they found that decreasing the concentration of the plasmid encoding NheC was, indeed, the key to successfully reconstituting active Nhe. Interestingly, increasing the expression level of *nheC* did not affect that of *nheA*; however, it adversely affected *nheB* expression levels, demonstrating the co-expression system’s complexity, even in the cell-free environment.

Expression levels of five genes (*vioA*–*E*) in the core metabolic pathway from tryptophan to violacein (Figure 3B) were modulated in a co-expression format in order to maximize the production of violacein, a potential anti-bacterial, anti-trypanocidal, anti-ulcerogenic, or anti-cancer drug [99], while keeping the accumulation of intermediates low [100]. In this work, the authors first demonstrated successful reconstruction of the metabolic pathway in a cell-free co-expression format. Initially, the pathway was reconstructed using equimolar concentrations of five PCR-amplified genes (4 nM each). Since the accumulation of intermediates suggested underperformance of the whole pathway, the enzyme expression levels were combinatorically modulated by supplementing the CFPS system with varying concentrations of corresponding genes. Even though exact expression levels were not presented, the three utilized levels of DNA input concentrations (low, medium, and high, at 2.5, 4, and 64.5 nM, respectively) were reported to have resulted in varying levels of enzyme expression and product yield. By measuring the byproduct titer as well as violacein, it was suggested that a high *vioC* and *vioD* DNA concentration resulted in a higher product yield while minimizing the accumulation of intermediates. Reconstruction of the violacein synthetic pathway was also undertaken by utilizing a freeze-dried CFPS system to co-express the five aforementioned genes [100]. Plasmids encoding each gene (*vioA*–*E*) were added to the CFPS system at varying concentrations (~0.1–10 nM). Interestingly, when all plasmids were added at the highest levels utilized in this experiment (40 ng/μL or 7–10 nM, depending on the construct), expression of *vioE*, and, thus, production of violacein, was not detected. Violacein production could be achieved by weakly expressing *vioA* and *vioB*, while keeping *vioC*–*E* levels comparatively high.

Using a synthetic metabolic pathway that utilizes a component of endogenous metabolisms in the cell lysate for CFPS also appears feasible. For example, a hybrid pathway including five cell-free synthesized enzymes, acetyl-CoA acetyltransferase (AtoB), hydroxybutyryl-CoA dehydrogenase (Hbd), crotonase (Crt), butyryl-CoA dehedrogenase (Ter), and bifunctional alcohol dehydrogenase (Adh) of the heterologous origins in the endogenous glycolytic pathway of the cell lysate for the conversion of glucose to *n*-butanol has been reported (Figure 3C). Initially, co-expressing all five genes in the CFPS system failed to yield *n*-butanol, due to the particularly low expression level of *adh*. However, increasing the concentration of the plasmid encoding an *adh* variant to occupy up to ~70% of the total DNA input resulted in the successful production of *n*-butanol [101]. Aside from bacteria, plant cells are also an effective platform for the production of secondary metabolites and heterologous proteins [102]. Recently, Buntru et al. established a synthetic metabolic pathway [103] that was a hybrid of the cell-free co-expressed enzymes and endogenous metabolic pathways, including the mevalonate pathway, of the CFPS system derived from plant based tobacco BY-2 cell lysate (Figure 3D). Lycopene, indigoidine, betanin, and betaxanthins could be successfully produced using this system [103]. In their work, plasmid molar concentrations remained constant, and only equimolar concentrations of plasmids were used to co-express multiple genes.

### 4.3. Attempts to Link Cell-Free Results to Microbial Cell Factories

As exemplified above, modulating the expression levels of multiple target genes in CFPS by using varying DNA concentrations may seem technically easy to accomplish. Even though these approaches are meaningful in terms of prototyping a synthetic metabolic pathway, the results are not expected to be directly transferable to cell factories, as modulating DNA concentrations in cells is not a suitable option. As is true of the cellular environment, the transcription and translation resources in the CFPS system are not limitless, and thus, the expression of one gene should affect expression of others. In principle, a certain gene can use most of the available transcription/translation resources, such as RNA polymerases and ribosomes, in a “winner takes all” manner. This has long been perceived empirically by CFPS researchers. For example, Keum et al. found that the co-expression of recombinant plasminogen activator and chloramphenicol acetyltransferase genes in a CFPS system resulted in the almost exclusive production of the former protein; in contrast, both genes were equally well-expressed when the CFPS reactions were conducted individually [104]. The authors could partially resolve the problem of disproportionate expression by introducing common nucleotide sequences, called the downstream box, throughout the genes to be co-expressed. By solving a series of ordinary differential equations describing transcription/translation processes, Marshall and Noireaux formulated the usage of transcriptional and translational resources in a CFPS system by single gene expression, using parameters such as promoter strength and gene length [105]. The concept that expression of one gene affects expression of others was also used to explain the effect of heterologous gene expression on cell growth [106]. Here, the authors investigated how CFPS could be utilized to better understand the expression competition occurring naturally in *E. coli* cells. In their model, a lumped parameter, γ, represented the cost of translating the gene of interest, and was regarded to depend on the nature of the protein being synthesized and the efficiency of translation. They utilized the CFPS system to estimate the cost of protein synthesis, and predicted the effect of expressing multiple genes on cell growth with a good correlation (R^2^ > 0.7) when additional burdens stemming from the implanted metabolic pathway were excluded.

In the prediction of in vivo protein expression costs described above, Borkowski et al. utilized bicistronic design (BCD) sequences to compare CFPS to cellular systems [106]. BCD is an expression cassette that comprises two cistrons under the same promoter. The first cistron has its own Shine–Dalgarno (SD) sequence and encodes 16 amino acid peptides. The coding region of the leader peptide is designed so that the second SD sequences have varying strength for the transition of the following peptide to the gene of interest (GOI), and initiation of gene expression for the second gene presumably occurs via a −1 frame shift from the stop codon (UAA) of the first cistron [107]. With proper selection of BCD sequences, one can expect defined levels of translation initiation regardless of the GOI in the second cistron. In a study conducted by [108], BCD sequences were utilized to correlate metabolite production with levels of enzyme expression, both in vitro and in vivo. To demonstrate the concept of iterative optimization of a synthetic metabolic pathway, 1,4-butanediol (BDO) synthesis from succinyl-coA in *E. coli* was selected (Figure 3E). Five genes were first expressed individually in a CFPS system, and the resultant enzymes in the crude reaction were mixed to successfully prototype the synthetic pathway. In this metabolic pathway reconstruction approach, the downstream enzyme(s) 4-hydroxybutyryl-coA reductase (Ald) and/or alcohol dehydrogenase (Adh) were thought to be limiting in the production of BDO, since the accumulation of γ-butyrolactone, which could form spontaneously from 4-hydroxybutyryl-coA, was observed to be much higher than BDO accumulation. To examine how the levels of two downstream enzymes, Ald and Adh, affected metabolic pathway performance, the authors separately modulated the expression levels of *ald* and *adh* genes in a cell-free co-expression format by using combinations of BCD sequences to differentially initiate translation. In this cell-free metabolic pathway expression approach, other enzymes in the pathway were also co-expressed in the CFPS system, but their levels were kept low by using BCD 22, which resulted in low-level expression regardless of the gene sequence in the second cistron [107]. With these methods, BDO production was found to linearly correlate with the level of *ald* expression, suggesting that Ald was the bottleneck enzyme in the production of BDO. Similar conclusions could be drawn from in vivo experiments using the same constructs as those that were involved in the cell-free experiments when the expression level of *adh* was kept at low or medium levels.

## 5. Conclusions

With the recent advancements in CFPS technology, including the use of automation, diversification of cellular sources for the system, and increased synthesis yield, it has become possible to apply bottom-up design approaches to virus synthetic biology and metabolic pathway engineering. The open nature of CFPS makes it ideal for building novel viruses (or VLPs) or synthetic pathways with great freedom. In bypassing cell culture for each design, CFPS can expedite the design–build–test routine of synthetic biology. For the same reason, CFPS would be more automation-friendly in biofoundry format, which can further expedite study. However, many synthetic designs from cell-free studies, even those that perform exceedingly well, will ultimately need to be verified in cell-based systems.

By utilizing a spectrum of BCD sequences rather than modulating DNA concentrations, the results for pathway performance can be extrapolated to an in vivo case to some extent [109]. It has been proven possible to prototype a synthetic metabolic pathway comprising many enzymes, and to pinpoint the bottleneck enzyme in the pathway. Additionally, the relative expression levels of cell-free synthesized enzymes have been reproduced in the cell system in a qualitative manner. However, more efforts are required to fine-map cell-free and in vivo systems in a quantitative manner. For example, overall expression levels of GOIs under the same BCD were higher in the cell than in the cell-free system [109], which potentially poses a burden on the cell to yield a pathway that performs more poorly than expected from the cell-free prototype. In fact, in the construction of the BDO pathway (Figure 3E), when *adh* expression was elevated to high levels using BCD 2, the BDO pathway performed poorly regardless of *ald* expression levels.

More examples of cell-free metabolic pathway expression in the context of specific chassis are desirable as well. Several hosts intended to serve as final microbial cell factories may have altered metabolisms, so that the devised pathways perform differently from the cell-free system derived from general *E. coli* hosts for recombinant protein production, such as BL21(DE3) and its derivatives. The dynamics of proteolysis should also be considered when extrapolating the results from BL21(DE3)-based cell-free system to other strains of *E. coli*.

## Figures and Tables

**Figure 1 microorganisms-10-02477-f001:**
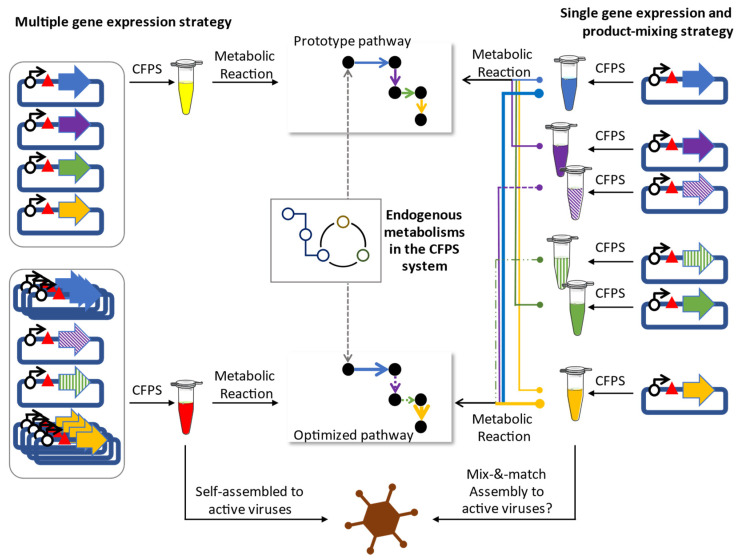
Cell-free multiple gene expression for viral particle synthesis or novel metabolic pathway construction. The one gene-one tube strategy is employed in typical CFPS approaches (**Right**). Genes for isoenzymes are shown in different shades in the same color. Even though this figure indicates that plasmids were used, amplified gene fragments can be used to prime the CFPS reaction. In this strategy, each product is mixed in varying ratios in search for an optimized metabolic pathway or successful viral particle reconstitutions. Large circles and thicker arrows are used to describe strengthened steps in the imaginary metabolic pathway. On the other hand, multiple genes are added to a single tube to accomplish the construction of a metabolic pathway or viral particles (**Left**). In this presentation, the concentration of each gene is manipulated to achieve varied ratios of all synthesized proteins, and, thus, optimal results. By completing the metabolic pathway in both strategies, endogenous metabolisms in the CFPS system can be utilized.

**Figure 2 microorganisms-10-02477-f002:**
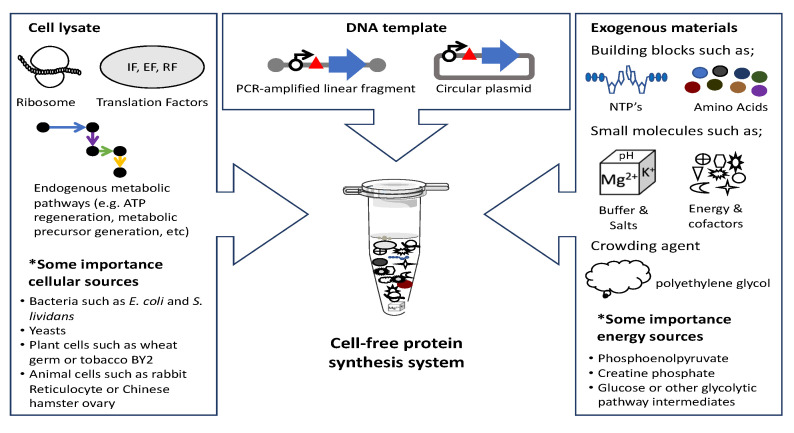
Typical cell-free protein synthesis system showing major components.

**Figure 3 microorganisms-10-02477-f003:**
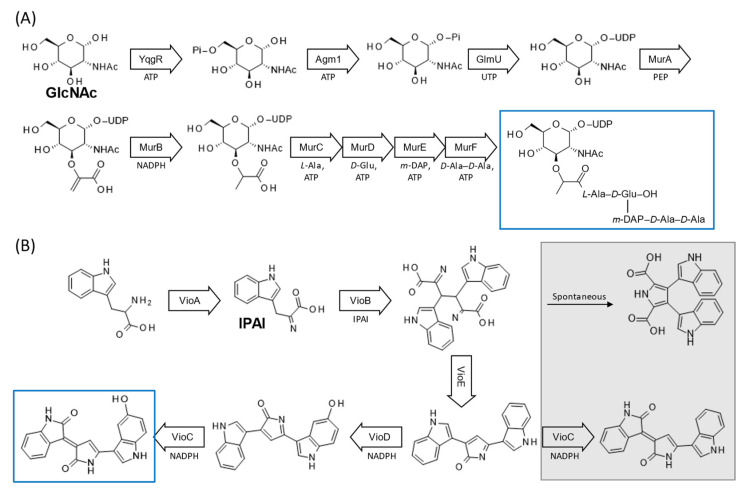
Metabolic pathways constructed by cell-free multiple gene expression. Single enzyme reactions are presented as solid arrows, with the enzyme’s name inside, and multiple step reactions involving multiple enzymes are shown as broken arrows. Key substrates of each reaction are shown beneath the arrow. Desired end products of the constructed pathways are shown in blue boxes, while unwanted side products and related reactions are shown in grey boxes. For simplicity, not all names of the intermediates are described. (**A**) Schematic diagram of the synthesis of UDP-N-acetylmuramyl pentapeptide from N-acetylglucosamine (**GlcNAc**). Pi, phosphate; PEP, phosphoenolpyruvate; m-DAP, meso-diaminopimelic acid. (**B**) Schematic diagram of a violacein biosynthetic pathway from tryptophan. IPAI, indole-3-pyrivic acid imine. (**C**) Schematic representation of a biosynthetic pathway to *n*-butanol from glucose using a hybrid pathway that consists of five cell-free expressed enzymes (indicated by solid arrows) and endogenous glycolytic pathway enzymes (shown in a box with a dashed line). (**D**) Schematic representation of a lycopene biosynthesis pathway from acetyl-CoA. The endogenous metabolic pathway, including the mevalonate pathway, is shown in a box with a dashed line. IPP, isopentenyl pyrophosphate; DMAPP, dimethylallyl diphosphate; GGPP, geranylgeranyl pyrophosphate. (**E**) The biosynthetic pathway to 1,4-butanediol (BDO) synthesis from succinyl-CoA. AcCoA, acetyl-CoA.

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
