# Peer review of "Multiple Gene Expression in Cell-Free Protein Synthesis Systems for Reconstructing Bacteriophages and Metabolic Pathways"

_microorganisms, 2022, doi:10.3390/microorganisms10122477_

Round 1

Reviewer 1 Report

The review by Purkayastha et al. of multi-gene expression in cell-free protein synthesis is well-written and provides useful information to many readers, so I judge that publication in Microorganisms is appropriate. However, the following modifications are required.

Since this review focuses on the synthesis of viral particles and metabolic pathway reconstruction, it should be added to the title.

In addition, for the viral particles, they describe bacteriophages and noninfectious virus-like particles, so the abstract should also include this information.

L191: 10^11 PFU/mL

Author Response

* The review by Purkayastha et al. of multi-gene expression in cell-free protein synthesis is well-written and provides useful information to many readers, so I judge that publication in Microorganisms is appropriate. However, the following modifications are required.

> We appreciate the reviewer's comments for improving the quality of the manuscript.

* Since this review focuses on the synthesis of viral particles and metabolic pathway reconstruction, it should be added to the title.

> We modified the title to "Multiple gene expression in cell-free protein synthesis sytems for reconstructing bacteriophages and metabolic pathways"

* In addition, for the viral particles, they describe bacteriophages and noninfectious virus-like particles, so the abstract should also include this information.

> The abstract was modified according to the reviewer's comment. The modification is shown in red in the modified version.

* L191: 10^11 PFU/mL

> Revised accordingly.

Reviewer 2 Report

The authors provided a holistic review on the recent advances using cell free systems on multiple gene expression. The review focuses on systems for the production of multiple gene products engineering, metabolic pathway engineering, and/or synthesis of organisms. The authors cited multiple examples on the optimization of the process. The review is well written and the two figures included help the audience to understand the process. This is very useful for synthetic biology community to continuously improve the process. It is suggested to the authors, however, the organization of the content of this review article could be improved. Subtitles are needed for each of the titles listed. It will be helpful to clearly state the argument at the beginning of each paragraph. Because it is a bit hard to follow all the details right now without an argument in the mind.

Also it will be helpful to clearly define cell free system at the beginning and add a figure to summarize the different type of cell free system that you mentioned in the paper. I know your Figure 1 is kind of help understanding but I think you can add a better image to summarize the cell free system mentioned in the manuscript. This will help the reader who is not too familiar with the cell free system to get on board.

Author Response

* The authors provided a holistic review on the recent advances using cell free systems on multiple gene expression. The review focuses on systems for the production of multiple gene products engineering, metabolic pathway engineering, and/or synthesis of organisms. The authors cited multiple examples on the optimization of the process. The review is well written and the two figures included help the audience to understand the process. This is very useful for synthetic biology community to continuously improve the process. It is suggested to the authors, however, the organization of the content of this review article could be improved. Subtitles are needed for each of the titles listed. It will be helpful to clearly state the argument at the beginning of each paragraph. Because it is a bit hard to follow all the details right now without an argument in the mind.

> We appreciate the reviewer for the comments. As was suggested, we added subtitles in the metabolic pathway reconstruction section. 

* Also it will be helpful to clearly define cell free system at the beginning and add a figure to summarize the different type of cell free system that you mentioned in the paper. I know your Figure 1 is kind of help understanding but I think you can add a better image to summarize the cell free system mentioned in the manuscript. This will help the reader who is not too familiar with the cell free system to get on board.

> As was suggested, we added one more figure describing the general idea about the cell-free protein synthesis system, emphasizing its key components. All the modifications made to the manuscript were colored in red.